# Chaotic behavior, sensitivity analysis and Jacobian elliptic function solution of M-fractional paraxial wave with Kerr law nonlinearity

Md. Mamunur Roshid[1,2]*, Mohammad Safi Ullah[3], M. M. Rahman[1], Harun-Or-Roshid[4]

1 Department of Mathematics, Bangladesh University of Engineering and Technology (BUET), Dhaka, Bangladeshig, 2 Department of Mathematics, Hamdard University Bangladesh, Munshiganj, Bangladesh, 3 Department of Mathematics, Comilla University, Comilla, Bangladesh, 4 Department of Mathematics, Sunamgonj Science and Technology University, Sunamganj, Bangladesh

* mamunmath307@gmail.com

**Data Availability Statement:** yes. We do not use any raw data. We use some simulation data. We upload the simulation data as the supporting file.

**Funding:** The author(s) received no specific funding for this work.

## Abstract

This study investigates the paraxial approximation of the M-fractional paraxial wave equation with Kerr law nonlinearity. The paraxial wave equation is most important to describe the propagation of waves under the paraxial approximation. This approximation assumes that the wavefronts are nearly parallel to the axis of propagation, allowing for simplifications that make the equation particularly useful in studying beam-like structures such as laser beams and optical solitons. The paraxial wave equation balances linear dispersion and nonlinear effects, capturing the essential dynamics of wave evolution in various media. It plays a crucial role in understanding phenomena like diffraction, focusing, and self-phase modulation in optical fibers. It substantially contributes to our comprehension of the special characteristics of optical soliton solutions and the dynamics of soliton in a variety of optical systems. We create a range of wave structures using the powerful extended Jacobian elliptic function expansion (EJEFE) method, including periodic waves, lump-periodic waves, periodic breather waves, kink-bell waves, kinky-periodic waves, anti-kinky-periodic waves, double-periodic waves, etc. These solutions have applications in wave dynamics in different optical systems and optical fibre. Furthermore, we investigate chaotic phenomena by analyzing the model qualitatively. We analyze phase portraits in detail for a range of parameter values to provide insights into the behavior of the system. We also investigate the sensitivity analysis for diverse parametric values of the perturbated coefficient. We may use various strategies, including time series and 3D and 2D phase patterns, to identify chaotic and quasi-periodic phenomena by providing an external periodic strength. The above discussion of the suggested method demonstrates adaptability and usefulness in resolving a broad spectrum of mathematics and physical difficulties, indicating its potential for generating such optical solutions.

**Competing interests:** The authors have declared that no competing interests exist.

## 1. Introduction

Optical soliton solutions are critically important in the field of fiber optics and telecommunications. These solitons allow for distortion-free, high-speed long-distance data transmission because they are stable, localized waves that hold their shape over extended distances. Their stability derives from a finely tuned dispersive and nonlinear interaction in the optic medium. This property makes optical solitons ideal for maintaining signal integrity in optical fibers, reducing the need for frequent signal regeneration and thereby enhancing the efficiency and cost-effectiveness of communication networks. Furthermore, optical solitons have applications in designing advanced photonic devices and systems, contributing to the development of ultra-fast optical switches and signal processors. The study of optical solitons also enriches our understanding of nonlinear wave dynamics, fostering innovations in both theoretical and applied physics. Thus, optical soliton solutions are fundamental in advancing modern optical technologies and communication systems [1–18], etc.

The study of the paraxial wave model is most significant to define the propagation of waves under the paraxial approximation. This model was introduced by R. Y. Chiao et al., in 1965. This approximation assumes that the wavefronts are nearly parallel to the axis of propagation, allowing for simplifications that make the equation particularly useful in studying beam-like structures such as laser beams and optical solitons. The paraxial wave equation balances linear dispersion and nonlinear effects, capturing the essential dynamics of wave evolution in various media. It plays a crucial role in understanding phenomena like diffraction, focusing, and self-phase modulation in optical fibers. Additionally, this model serves as a cornerstone for developing advanced photonic devices and systems, including high-speed communication networks and ultrafast optical processors. The study of paraxial wave equations not only advances theoretical physics but also drives technological innovations in fields ranging from telecommunications to medical imaging. In this study, the time M-fractional paraxial wave (tM-fPW) model is deliberated as [19, 20]:

$$iU_x + \frac{\varepsilon_1}{2} {}_{\kappa}D_{M,t}^{\sigma,n} U + \frac{\varepsilon_2}{2} U_{zz} + \varepsilon_3 |U|^2 U = 0, \tag{1}$$

where $U = U(x,z,t)$ and $\varepsilon_1, \varepsilon_2, \varepsilon_3$ are real constant and $\varepsilon_1$ is effect of dispersal, $\varepsilon_2$ is effect of diffraction and $\varepsilon_3$ is effect of kerr non-linearity. The term $iU_x$ captures the forward propagation of the wave and its phase evolution in the $x$-direction. The terms $\frac{\varepsilon_1}{2} {}_{\kappa}D_{M,t}^{\sigma,n} U$ and $\frac{\varepsilon_2}{2} U_{zz}$ represent the effects of dispersion or diffraction in the $z$ and $t$ directions, respectively. These terms cause the wave to spread out as it propagates. The term $\varepsilon_3 |U|^2 U$ describes the nonlinear interaction of the wave with itself. This term leads to effects such as self-focusing or defocusing, depending on the sign of $\varepsilon_3$.

Determining various optical wave structures for PW models has been accomplished recently through a variety of techniques. For example, W. Gao implemented the MAE technique [20] to evaluate the instability and acquire some soliton for the PW model. T. Rasool used the Sardar-sub equation method [21] to time the M-fractional PW equation and acquire some periodic solutions. Hamood Ur Rehman adopted the $\varphi^6$-expansion technique [22] to find the precise solution of the PW equation. M. Arshad used ISE, MEDA, and Exp($-\Phi(H)$)-expansion approach [23] to solve the PW model. M.M. Roshid implemented a unified scheme [24] to time fractional PW model. N. Ullah implemented Kudryashov and Tanh methods to find the precise solutions of the PW dynamic model [25], Extrapolation of Richardson was implemented by V. R. Chinni to investigate the precise solution of the paraxial wave equation [26]. Hamood Ur Rehman uses the SS-E approach [27] to control the propagation of monochromatic optical beams. Kashif Ali integrated optical soliton solution by extended trial

equation approach [28], Manaf et al. investigated the behaviour of optical self-control soliton by utilizing the ET and MET method [29], and so on. The present work aims to examine the optical wave patterns of the fM-fPW equation. By implementing the EJEFE method, the optical solution is explored, and also check the influence of fractional parameters for $\sigma = 0.1, 0.5, 0.9$ etc. Additionally, we also investigate the chaotic nature through a qualitative analysis of the model at the first time. For an assortment of parameter values, we conduct a thorough analysis of phase profiles to shed light on the system's behavior. We also investigate the sensitivity analysis for diverse parametric values of the protuberate coefficient. By introducing an external periodic strength, With the use of several techniques, including time series and 2D and 3D phase structure, we can distinguish between chaotic and quasi-periodic phenomena.

## 2. Methodology

### 2.1 Preliminary of M-fractional derivative

#### 2.1.1. Definition

Given a function u:$[0,\infty)\rightarrow\Re$ and an order $\sigma$, the truncated M-fractional derivative is defined as follows [30–34]:

$$_{\kappa}D_{M,t}^{\sigma,\phi}\, u(t) = \lim_{\epsilon\to 0}\frac{u(t_{\kappa}E_{\phi}(\epsilon t^{-\sigma})) - u(t)}{\epsilon}, t > 0, \phi > 0.$$

Here, $E_{\phi}(t)$ is a truncated Mittag-Leffler function of one parameter, defined as [32], and taking values in the interval (0,1).

$$_{\kappa}E_{\phi}(t) = \sum_{n=0}^{\kappa}\frac{t^{n}}{\Gamma(\phi n + 1)}.$$

#### 2.1.2. Features

Now we explain the properties truncated M-fractional operator $_{\kappa}D_{M,t}^{\sigma,\phi}$, here $\phi$ can represent a phase, angle, or additional parameter that modifies the operator, $\sigma$ is often associated with the order of the fractional derivative. It specifies the degree of the fractional operation, where $0<\sigma<1$ usually corresponds to a fractional derivative, with $\sigma = 1$ recovering the standard first-order derivative, $t$ represents the temporal variable in the system being studied, $M$ usually indicates the truncation point or order, $D$ represents the fractional derivative operator, the parameter $\kappa$ usually represents a scaling factor or a characteristic constant associated with the operator. Considered that $0<\sigma<1$, and $l,m$ are arbitrary constants. Let $u,v$ be functions of time that are $\sigma$-differentiable at a point $\phi>0$. Then,

(a) $_{\kappa}D_{M,t}^{\sigma,\phi}(lu + mv) = l_{\kappa}D_{M,t}^{\sigma,\phi}(u) + m D_{M,t}^{\sigma,\phi}(v),$

(b) $_{\kappa}D_{M,t}^{\sigma,\phi}(uv) = u_{\kappa}D_{M,t}^{\sigma,\phi}(v) + v_{\kappa}D_{M,t}^{\sigma,\phi}(u),$

(c) $_{\kappa}D_{M,t}^{\sigma,\phi}\left(\frac{u}{v}\right) = v_{\kappa}D_{M,t}^{\sigma,\phi}(u) - u\frac{_{\kappa}D_{M,t}^{\sigma,\phi}(v)}{v^2},$

(d) $_{\kappa}D_{M,t}^{\sigma,\phi}(t^{\chi}) = \phi t^{\chi-\sigma}, \phi \in \Re,$

(e) $_{\kappa}D_{M,t}^{\sigma,\phi}(c) = 0$; $c$ is an arbitrary constant.

(f) $_{\kappa}D_{M,t}^{\sigma,\phi}(u \circ v)(t) = u'(v)_{\kappa}D_{M,t}^{\sigma,\phi}v(t)$; If $u$, is differentiable at $v$.

(g) $_{\kappa}D_{M,t}^{\sigma,\phi}u(t) = \frac{t^{1-\sigma}}{\Gamma(\phi+1)}\frac{du}{dt}$; If $u$, is differentiable.

## 2.2 The extended Jacobian elliptic function expansion approach (EJEFE)

The subsequent nonlinear ordinary differential equation in Eq (2) can be used to solve a number of NLPDEs in computational physics.

$$\mathfrak{H}_1 H'' + \mathfrak{H}_2 H^3 + \mathfrak{H}_3 H = 0. \tag{2}$$

The closed-form waveform patterns of the NPDE solutions are offered by the EJEFE method [35, 36] in the following form.

$H = q_1 + \sum_{k=1}^{M} q_{1+k}(sn(\varphi))^k + \sum_{k=1}^{M} p_k(cn(\varphi))^k$; here $M$ is the balance number between the highest derivative and nonlinear terms and $q_1, q_{1+k}$ and $p_k$ are arbitrary constants.

Balancing $H''$ and $H^3$ for Eq (2), yields the balance number is $M = 1$. The trial solution of Eq (2) is:

$$H = q_1 + q_2 sn(\varphi) + p_1 cn(\varphi). \tag{3}$$

Here $q_1, q_2$ and $p_1$ are arbitrary constants.

The integration form of Eq (2) is

$$H' = q_2 cn(\varphi)dn(\varphi) - p_1 sn(\varphi)dn(\varphi). \tag{4}$$

$$\left.\begin{array}{c} H'' = -m^2 sn(\varphi)q_2 + 2q_2 sn(\varphi)^3 m^2 \\ +2m^2 sn(\varphi)^2 cn(\varphi)p_1 - q_2 sn(\varphi) - p_1 cn(\varphi) \end{array}\right\}. \tag{5}$$

Superseding Eqs (3)–(5) into Eq (2) and set the quantity of $sn^3$, $sn^3 cn$, $sn^2$, $sncn$, $sn$, $cn$, $sn^0$ with zero, arrangement algebraic equations. Resolving this system, crops:

**Family 01**: $q_1 = 0, q_2 = \pm\sqrt{-\frac{2\mathfrak{H}_1}{\mathfrak{H}_2}}m, p_1 = 0, \mathfrak{H}_3 = \mathfrak{H}_1(1 + m^2).$

$$H = \pm\sqrt{-\frac{2\mathfrak{H}_1}{\mathfrak{H}_2}}m sn(\varphi).$$

$$H = \pm\sqrt{-\frac{2\mathfrak{H}_1}{\mathfrak{H}_2}}tanh(\varphi); \ [m \to 1].$$

**Family 02**: $q_1 = 0, q_2 = \pm\sqrt{-\frac{\mathfrak{H}_1}{2\mathfrak{H}_2}}m, p_1 = -\sqrt{\frac{\mathfrak{H}_1}{2\mathfrak{H}_2}}, \mathfrak{H}_3 = \frac{1}{2}\mathfrak{H}_1(2 - m^2).$

$$H = \pm\sqrt{-\frac{2\mathfrak{H}_1}{\mathfrak{H}_2}}m sn(\varphi) - \sqrt{\frac{\mathfrak{H}_1}{2\mathfrak{H}_2}}m cn(\varphi).$$

$$H = \pm\sqrt{-\frac{2\mathfrak{H}_1}{\mathfrak{H}_2}}tanh(\varphi) - \sqrt{\frac{\mathfrak{H}_1}{2\mathfrak{H}_2}}sech(\varphi); \ [m \to 1].$$

**Family 03**: $q_1 = 0, q_2 = \pm\sqrt{-\frac{2\mathfrak{H}_1}{\mathfrak{H}_2}}m, p_1 = -\sqrt{\frac{\mathfrak{H}_1}{2\mathfrak{H}_2}}, \mathfrak{H}_3 = \frac{1}{2}\mathfrak{H}_1(2 - m^2).$

$$H = \pm\sqrt{-\frac{2\mathfrak{H}_1}{\mathfrak{H}_2}}msn(\varphi) - \sqrt{\frac{\mathfrak{H}_1}{2\mathfrak{H}_2}}mcn(\varphi).$$

$$H = \pm\sqrt{-\frac{2\mathfrak{H}_1}{\mathfrak{H}_2}}tanh(\varphi) - \sqrt{\frac{\mathfrak{H}_1}{2\mathfrak{H}_2}}sech(\varphi); \ [m \to 1].$$

**Family 04**: $q_1 = 0, q_2 = 0, p_1 = \sqrt{\frac{\mathfrak{H}_1}{2\mathfrak{H}_2}}m, \mathfrak{H}_3 = \frac{1}{2}\mathfrak{H}_1(1 - 2m^2).$

$$H = \sqrt{\frac{\mathfrak{H}_1}{2\mathfrak{H}_2}}mcn(\varphi).$$

$$H = \sqrt{\frac{\mathfrak{H}_1}{2\mathfrak{H}_2}}sech(\varphi); \ [m \to 1].$$

## 3. Formation of closed-form optical soliton solutions of the tM-fPW model

In this section, we examine the analytic explanations of the fractional PW model by implementing EJEFE techniques. The form of the pulse depicted is considered as, $U(x,z,t) = U(\eta)e^{i\varphi}$ in Eq (1)

$$\eta = \left(g_1 x + g_2 z + \omega\frac{\Gamma(n+1)}{\sigma}t^\sigma\right), \varphi = \left(h_1 x + h_2 z + \tau\frac{\Gamma(n+1)}{\sigma}t^\sigma\right) + \delta. \tag{6}$$

By inserting Eq (6) in Eq (1) and unraveling into imaginary and real parts,
We gain,

$$(\varepsilon_2 g_1^2 + \varepsilon_1\omega^2)U" - 2\varepsilon_3 U^3 - (\varepsilon_1\tau^2 + 2h_2 + \varepsilon_2 h_1^2)U = 0. \tag{7}$$

and,

$$(2\varepsilon_1\tau\omega + 2\varepsilon_2 g_1 h_1 + 2g_2)U' = 0. \tag{8}$$

As $U'{\neq}0$, so Eq (8) becomes,

$$g_2 = -\varepsilon_1\tau\omega - \varepsilon_2 g_1 h_1.$$

The closed-form waveform patterns of the tM-fPW equation are offered by the proposed method.

$$\mathfrak{H}_1 U'' + \mathfrak{H}_2 U^3 + \mathfrak{H}_3 U = 0, \tag{9}$$

where, $\mathfrak{H}_1 = (\varepsilon_2 g_1^2 + \varepsilon_1\omega^2), \mathfrak{H}_2 = -2\varepsilon_3$, and $\mathfrak{H}_3 = -(\varepsilon_1\tau^2 + 2h_2 + \varepsilon_2 h_1^2).$

Balancing $H''$ and $H^3$ for Eq (9), yields the balance number $M = 1$. The trial solution of Eq (9) is:

$$U = q_1 + q_2 sn(\varphi) + p_1 cn(\varphi). \tag{10}$$

Here $q_1, q_2$ and $p_1$ are arbitrary constants.

$$U' = q_2 cn(\varphi)dn(\varphi) - p_1 sn(\varphi)dn(\varphi). \tag{11}$$

$$\left.\begin{array}{c} U'' = -m^2 sn(\varphi)q_2 + 2q_2 sn(\varphi)^3 m^2 \\ +2m^2 sn(\varphi)^2 cn(\varphi)p_1 - q_2 sn(\varphi) - p_1 cn(\varphi) \end{array}\right\}. \tag{12}$$

Superseding Eqs (10)–(12) into Eq (9) and set the quantity of $sn^3$, $sn^3\ cn$, $sn^2$, $sncn$, $sn$, $cn$, $sn^0$ with zero, arrangement algebraic equations. Resolving this system, crops:

**Family 01:** $q_1 = 0, q_2 = \pm\sqrt{\frac{2(\varepsilon_2 g_1^2 + \varepsilon_1 \omega^2)}{2\varepsilon_3}}m, p_1 = 0, \omega = \sqrt{\frac{-(\varepsilon_2 g_1^2 + \varepsilon_1 \tau^2 + 2h_2 + \varepsilon_2 h_1^2)}{2\varepsilon_1}}.$

$$R = \pm\left(\sqrt{-\frac{2\mathfrak{H}_1}{\mathfrak{H}_2}}tanh(\varphi)\right)e^{i\varphi}; \ [m \to 1], \tag{13}$$

where $\eta = \left(g_1 x - \left(\varepsilon_1 \tau \sqrt{\frac{-(\varepsilon_2 g_1^2 + \varepsilon_1 \tau^2 + 2h_2 + \varepsilon_2 h_1^2)}{2\varepsilon_1}} + \varepsilon_2 g_1 h_1\right)z + \sqrt{\frac{-(\varepsilon_2 g_1^2 + \varepsilon_1 \tau^2 + 2h_2 + \varepsilon_2 h_1^2)}{2\varepsilon_1}}\frac{\Gamma(n+1)}{\sigma}t^\sigma\right),$

$\varphi = \left(h_1 x + h_2 z + \tau \frac{\Gamma(n+1)}{\sigma}t^\sigma\right) + \delta.$

**Family 02:** $q_1 = 0, q_2 = \pm\sqrt{\frac{(\varepsilon_2 g_1^2 + \varepsilon_1 \omega^2)}{4\varepsilon_3}}m, p_1 = \sqrt{\frac{(\varepsilon_2 g_1^2 + \varepsilon_1 \omega^2)}{-4\varepsilon_3}}, g_1 = \sqrt{\frac{-2(\varepsilon_1 \tau^2 + 2h_2 + \varepsilon_2 h_1^2 + \varepsilon_1 \omega^2)}{\varepsilon_2}}.$

$$R = \left(\pm\sqrt{-\frac{2L_1}{L_2}}tanh(\varphi) + \sqrt{\frac{(\varepsilon_2 g_1^2 + \varepsilon_1 \omega^2)}{-4\varepsilon_3}}sech(\varphi)\right)e^{i\varphi}; \ [m \to 1], \tag{14}$$

where $\eta = \left(\sqrt{\frac{-2(\varepsilon_1 \tau^2 + 2h_2 + \varepsilon_2 h_1^2 + \varepsilon_1 \omega^2)}{\varepsilon_2}}x - \left(\varepsilon_1 \tau\omega + \varepsilon_2\sqrt{\frac{-2(\varepsilon_1 \tau^2 + 2h_2 + \varepsilon_2 h_1^2 + \varepsilon_1 \omega^2)}{\varepsilon_2}}h_1\right)z + \omega\frac{\Gamma(n+1)}{\sigma}t^\sigma\right),$

$\varphi = \left(h_1 x + h_2 z + \tau \frac{\Gamma(n+1)}{\sigma}t^\sigma\right) + \delta.$

**Family03:** $q_1 = 0, q_2 = \pm\sqrt{\frac{(\varepsilon_2 g_1^2 + \varepsilon_1 \omega^2)}{\varepsilon_3}}m, p_1 = -\sqrt{\frac{(\varepsilon_2 g_1^2 + \varepsilon_1 \omega^2)}{-4\varepsilon_3}}, g_1 = \sqrt{\frac{-2(\varepsilon_1 \tau^2 + 2h_2 + \varepsilon_2 h_1^2 + \varepsilon_1 \omega^2)}{\varepsilon_2}}.$

$$R = \left(\pm\sqrt{\frac{(\varepsilon_2 g_1^2 + \varepsilon_1 \omega^2)}{\varepsilon_3}}tanh(\varphi) - \sqrt{\frac{(\varepsilon_2 g_1^2 + \varepsilon_1 \omega^2)}{-4\varepsilon_3}}sech(\varphi)\right)e^{i\varphi}; \ [m \to 1], \tag{15}$$

where $\eta = \left(\sqrt{\frac{-2(\varepsilon_1 \tau^2 + 2h_2 + \varepsilon_2 h_1^2 + \varepsilon_1 \omega^2)}{\varepsilon_2}}x - \left(\varepsilon_1 \tau\omega + \varepsilon_2\sqrt{\frac{-2(\varepsilon_1 \tau^2 + 2h_2 + \varepsilon_2 h_1^2 + \varepsilon_1 \omega^2)}{\varepsilon_2}}h_1\right)z + \omega\frac{\Gamma(n+1)}{\sigma}t^\sigma\right),$

$\varphi = \left(h_1 x + h_2 z + \tau \frac{\Gamma(n+1)}{\sigma}t^\sigma\right) + \delta.$

**Family 04:** $q_1 = 0, q_2 = 0, p_1 = \sqrt{\frac{(\varepsilon_2 g_1^2 + \varepsilon_1 \omega^2)}{-4\varepsilon_3}}m, \tau = \sqrt{\frac{-2(2h_2 + \varepsilon_2 h_1^2) + (\varepsilon_2 g_1^2 + \varepsilon_1 \omega^2)}{\varepsilon_1}}.$

$$R = \sqrt{\frac{(\varepsilon_2 g_1^2 + \varepsilon_1 \omega^2)}{-4\varepsilon_3}}sech(\varphi)e^{i\varphi}; \ [m \to 1], \tag{16}$$

where $\eta = \left(g_1 x - (\varepsilon_1 \tau\omega + \varepsilon_2 g_1 h_1)z + \omega\frac{\Gamma(n+1)}{\sigma}t^\sigma\right), \varphi = \left(h_1 x + h_2 z + \tau\frac{\Gamma(n+1)}{\sigma}t^\sigma\right) + \delta.$

## 4. Chaotic nature

This section investigates the periodic, quasiperiodic, and chaotic dynamics of system Eq (7) by adding an external, superficial component [37, 38]. To obtain these dynamics, consider the

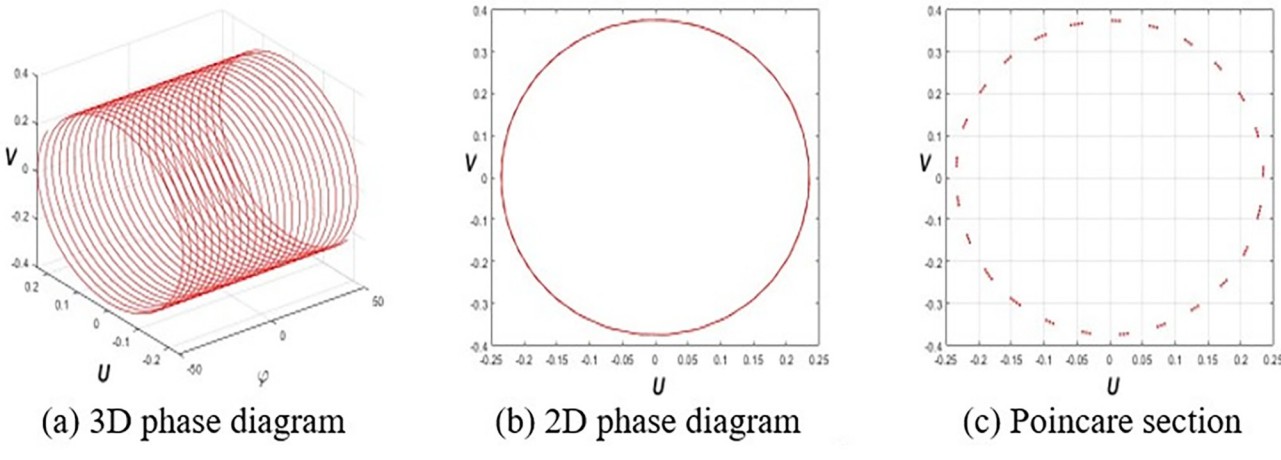

**Fig 1. Periodic nature of Eq (17) for $E = 0$, and $(A(0),B(0)) = (0.2,0.2)$.**

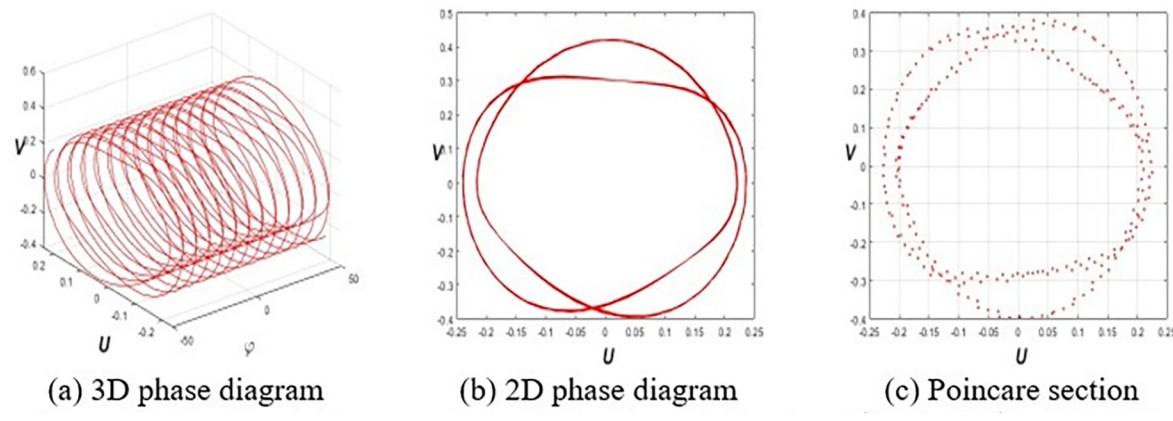

**Fig 2. Quasi-periodic nature of Eq (17) for $E = 0.2$, $F = 4$ and $(E(0),F(0)) = (0.2,0.2)$.**

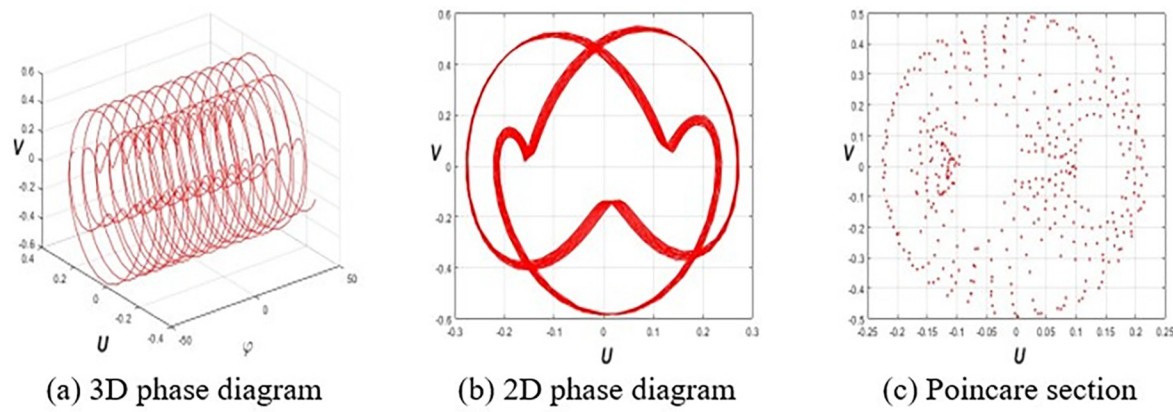

**Fig 3. Qusai-periodic nature of Eq (17) for $E = 75$, $F = 4$ and $(E(0),F(0)) = (0.2,0.2)$.**

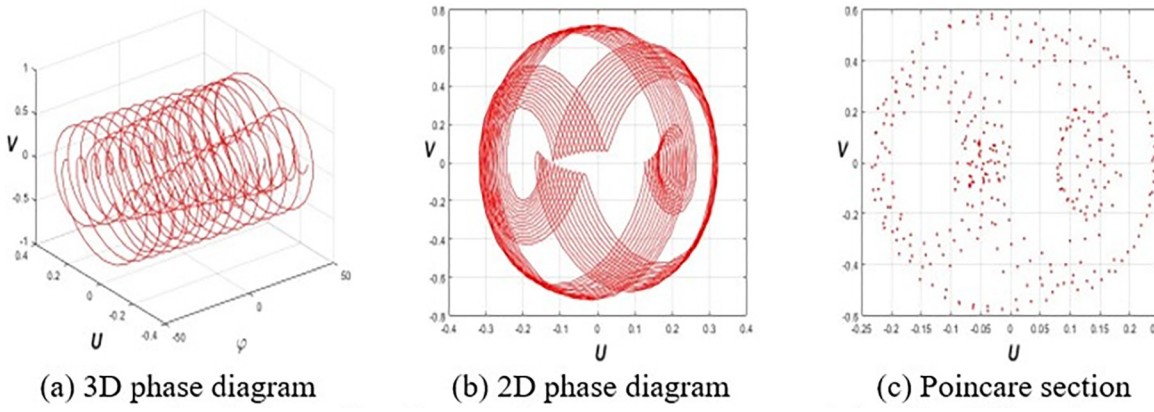

**(a) 3D phase diagram** **(b) 2D phase diagram** **(c) Poincare section**

**Fig 4. Chaotic dynamics of Eq (17) for *E* = 1.1, *F* = 4 and (*E*(0),*F*(0)) = (0.2,0.2).**

system Eq (7) in the subsequent form:

$$
\begin{cases}
\dfrac{dU}{d\eta} = V, \\[2mm]
\dfrac{dV}{d\eta} = \dfrac{(\varepsilon_1\tau^2 + \varepsilon_2 h_1^2 + 2h_2)U}{(\varepsilon_1\omega^2 + \varepsilon_2 g_1^2)} + \dfrac{2\varepsilon_3 U^3}{(\varepsilon_1\omega^2 + \varepsilon_2 g_1^2)} + E\cos(F\varphi),
\end{cases}
\tag{17}
$$

where $E$ and $F$ represent the strength and frequency of the perturbation term $E\cos(F\varphi)$, respectively. To reach our destination, we employ 3D phase portraits, 2D phase portraits, and Poincaré sections for the parameters $\varepsilon_1 = \varepsilon_2 = h_1 = g_1 = \omega = \tau = 1, \varepsilon_3 = -1.5$, and $h_2 = -3.5$ with initial value $(U(0), V(0)) = (0.2, 0.2)$. **Fig 1** shows the periodic nature of system Eq (17) for $E = 0$. **Figs 2** and **3** signify the quasiperiodic pattern of system Eq (17) for $E = 0.2$, $F = 4$ and $E = 0.75$, $F = 4$, respectively. **Figs 4** and **5** represent the chaotic behavior of system Eq (17) for $E = 1.1$, $F = 4$ and $E = 3.2$, $F = 4$, respectively.

It is important to note that the chaotic behavior observed in the solutions to Eq (17) arises due to the introduction of an oscillatory perturbation to the system. In the traditional sense of chaos theory, chaos is marked by being sensitive to initial conditions rather than changes in the system itself. This doesn't necessarily mean that the original system described by Eq (7) or Eq (1) is chaotic. This distinction aligns with the definition of chaos introduced by Edward N. Lorenz and is crucial for accurately characterizing the system's dynamics [39].

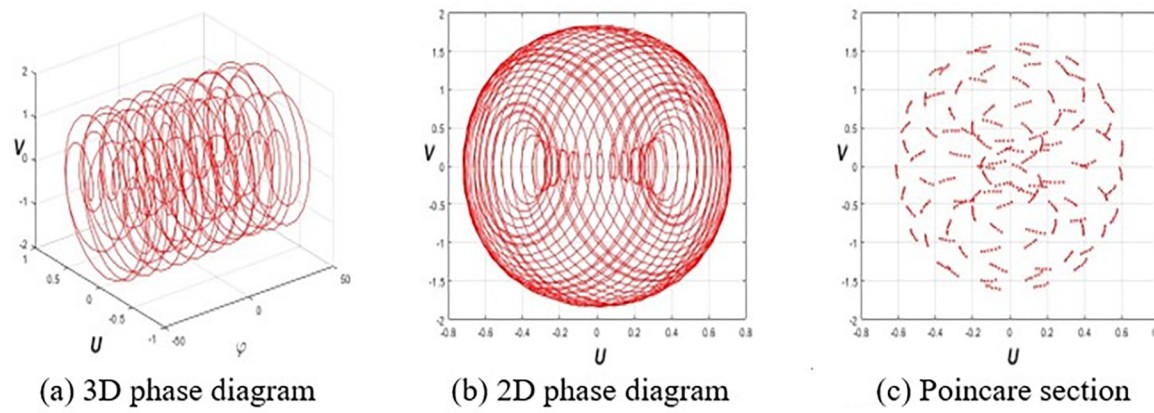

**(a) 3D phase diagram** **(b) 2D phase diagram** **(c) Poincare section**

**Fig 5. Chaotic dynamics of Eq (17) for *E* = 3.2, *F* = 4 and (*E*(0),*F*(0)) = (0.2,0.2).**

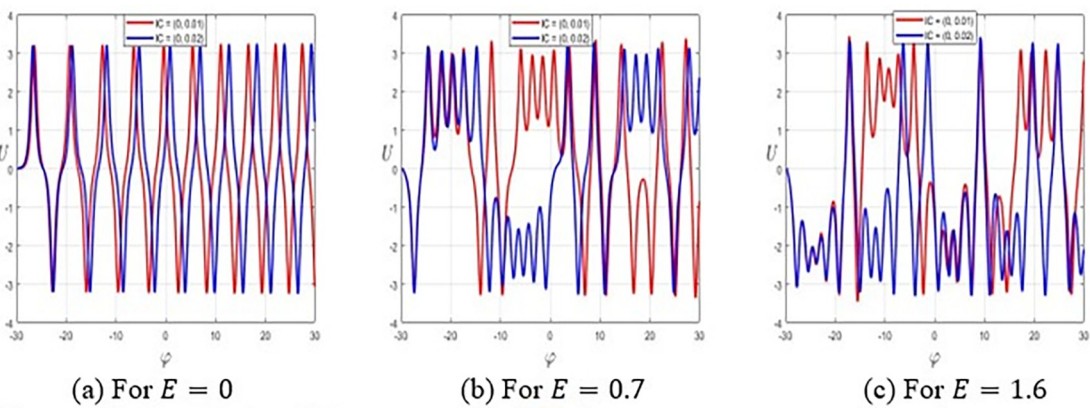

**Fig 6. Sensitivity of the system Eq (17) for $\varepsilon_1 = -3, \varepsilon_2 = \varepsilon_3 = h_1 = h_2 = g_1 = \omega = 1, \tau = 2.1$, and F = 2.4.**

**Fig 7. 3D and 2D profile of the solution Eq (13) for the values $\theta = -0.1, \varepsilon_3 = -1, z = 1, \varepsilon_2 = 0.2, \tau = 1, g_1 = -0.5, h_1 = -0.5, h_2 = -0.01, \varepsilon_1 = -1, n = 1.5.$**

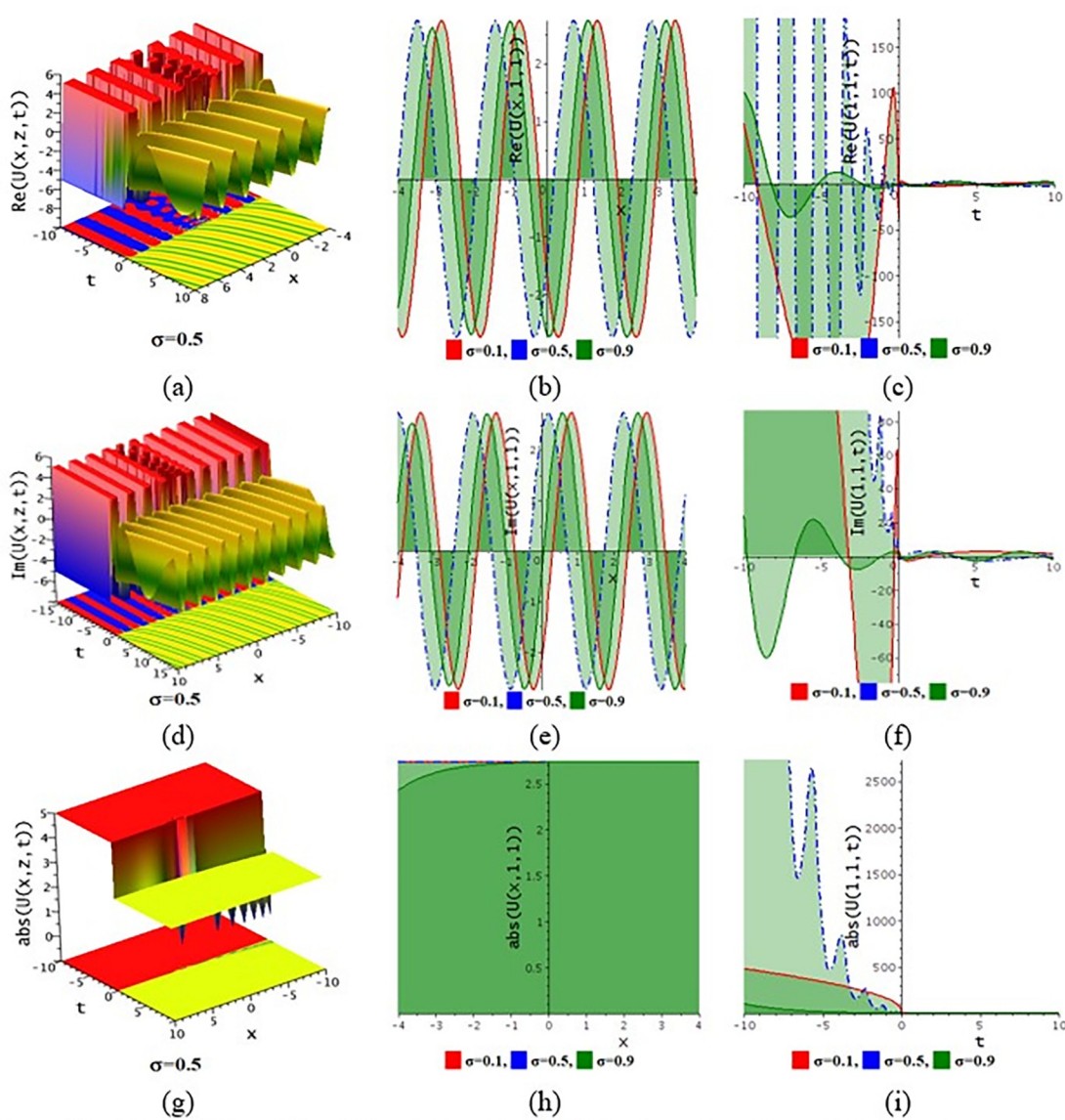

**Fig 8. 3D and 2D profile of the solution Eq (13) for the values** $\theta = -0.1, \varepsilon_3 = 1, z = 1, \varepsilon_2 = -0.2, \tau = 1, g_1 = 0.5, h_1 = -3, h_2 = -1.5, \varepsilon_1 = 1, n = 1.5$.

## 5. Sensitivity analysis

This portion investigates the model's sensitivity with different initial settings [40, 41]. To measure the sensitivity of the governing model, we use two sets of initial conditions: $(U,V) = (0,0.01)$ and $(U,V) = (0,0.02)$, plotted by the red and blue curves, sequentially. When the disturbed term is not involved in the system (17), meaning $E = 0$, **Fig 6A** shows that the system displays low sensitivity to the initial values. However, as the amplitude of the disturbed term rises, **Fig 6B and 6C** reveal that system (17) becomes highly sensitive to small changes in initial values at $E = 0.7$ and $E = 1.6$, separately. This behavior proves the chaotic attitude of the suggested nonlinear problem.

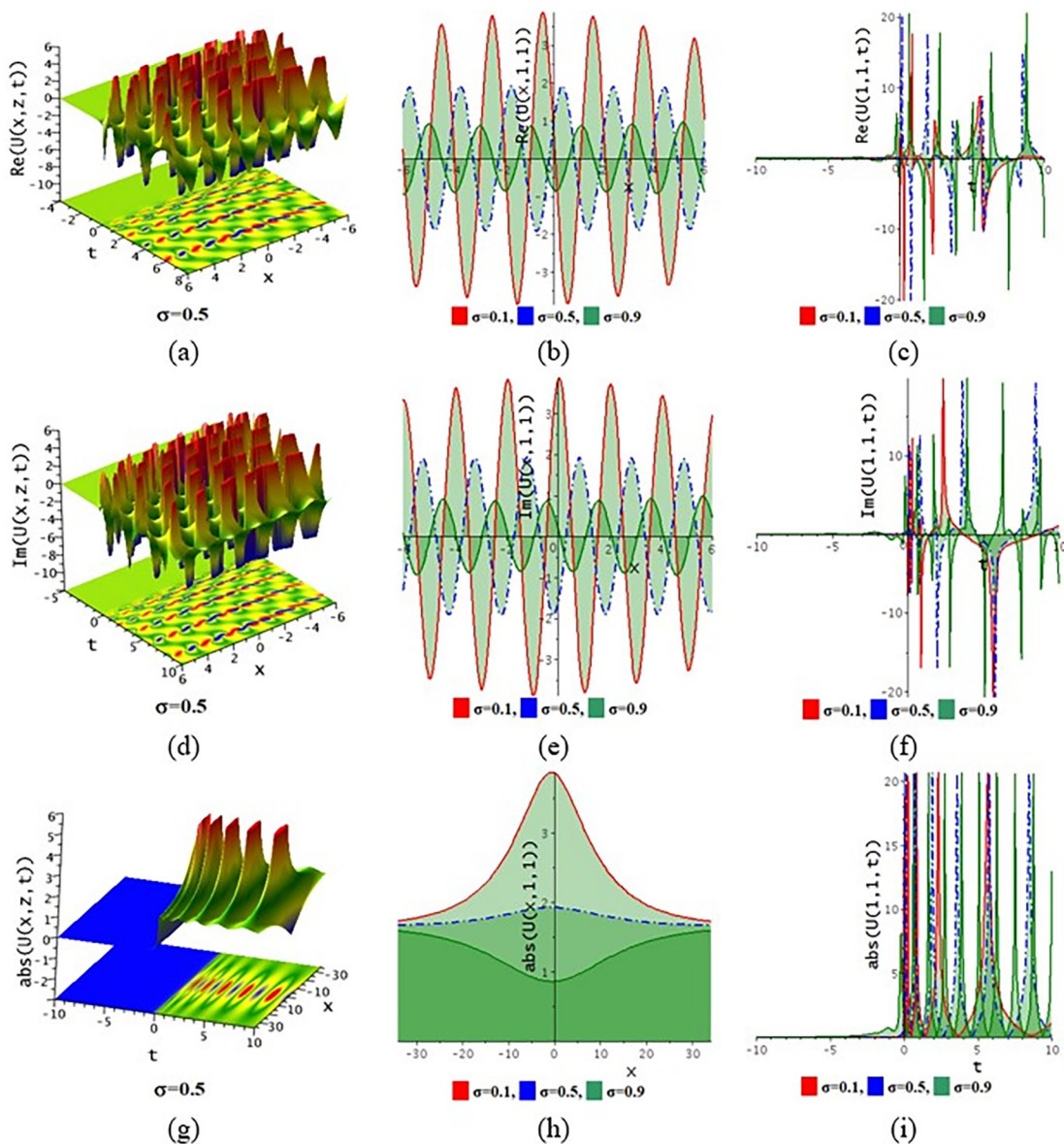

**Fig 9. 3D and 2D profile of the solution Eq (14) for the values** $\theta = 0.1, \varepsilon_3 = 2, \varepsilon_2 = 0.1, \tau = -2, g_1 = 0.1, h_1 = 3, h_2 = 3, \varepsilon_1 = 1, n = 1.5.$

## 6. Numerical discussion and graphical representation

The paraxial wave model with Kerr law nonlinearity is a partial differential equation that describes optical wave propagation in the paraxial approximation. This equation comes from the Maxwell equations, which describe the movement of the electromagnetic spectrum. An essential tool in optics for examining optical wave propagation in the paraxial approximation is the paraxial wave equation. Through the use of integrating approaches, scientists and engineers may find answers and forecast how optical systems will behave in a variety of applications. The Kerr law nonlinearity is more important in optics as it describes the intensity-dependent change in the refractive index of a material. This phenomenon, where the refractive

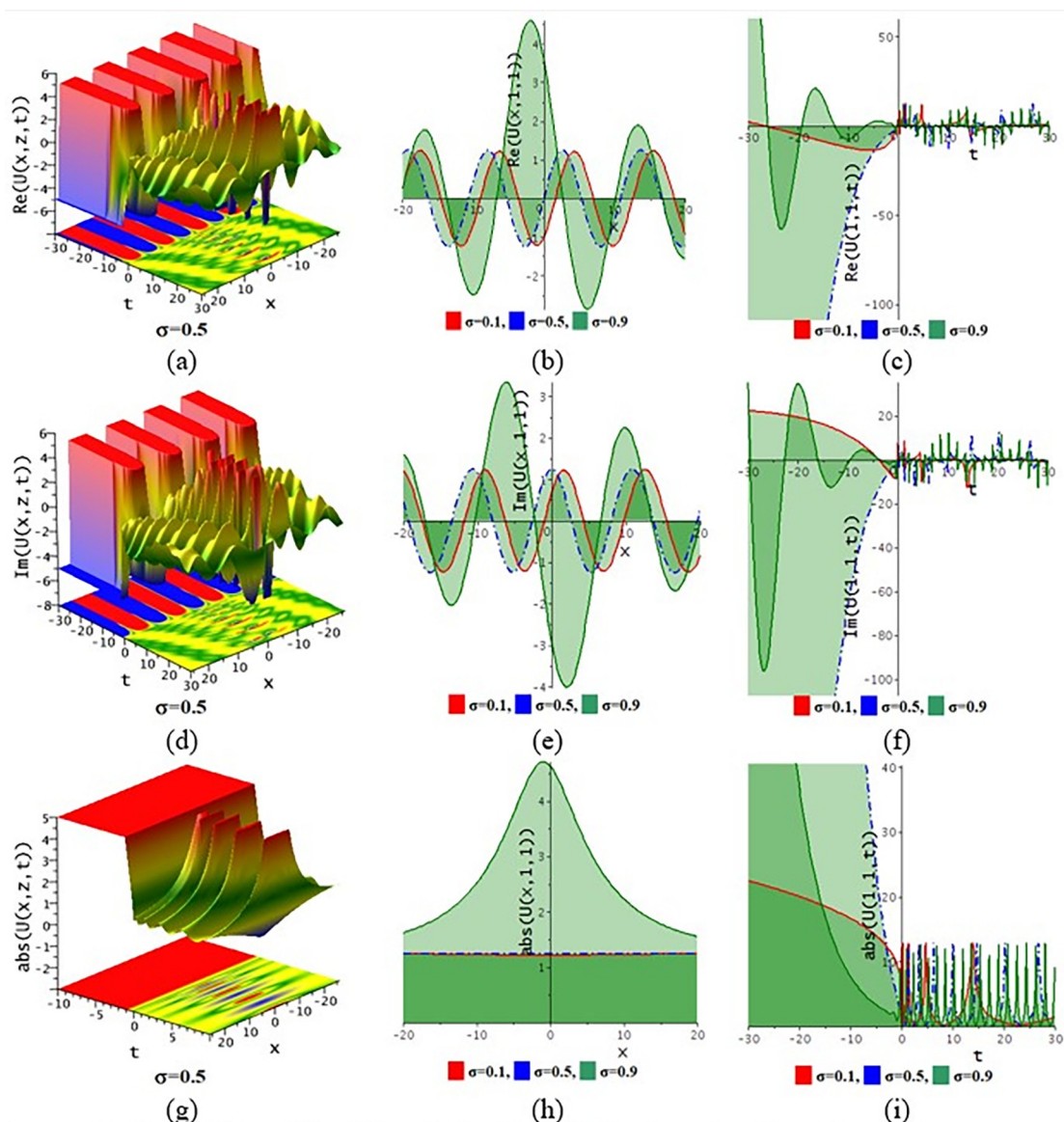

**Fig 10. 3D and 2D profile of the solution Eq (14) for the values** $\theta = 0.1, z = 1, \varepsilon_3 = 2, \varepsilon_2 = 0.1, \tau = 0.5, g_1 = 0.1, h_1 = -0.5, h_2 = 3, \varepsilon_1 = 1, n = 1.5.$

index increases with the light intensity, is crucial for various applications such as optical switching, pulse shaping in fiber optics, and in the development of nonlinear optical devices. The Kerr effect enables the manipulation of light within photonic circuits, contributing to advancements in optical communication technologies. Its role in enhancing the performance and functionality of optical systems highlights its importance in modern photonics and related fields. In this subdivision, We go over the behaviors of the photonic solutions for the M-fractional Paraxial wave model using the EJEFE method's numerical, 3D, and 2D graphical forms. By using the EJEFE method, different types of optical wave patterns are investigated in Figs 7 to 12 such as periodic waves, the interaction of periodic waves and lump waves, periodic breather waves, the interaction of kink and bell waves, kinky periodic waves, double periodic

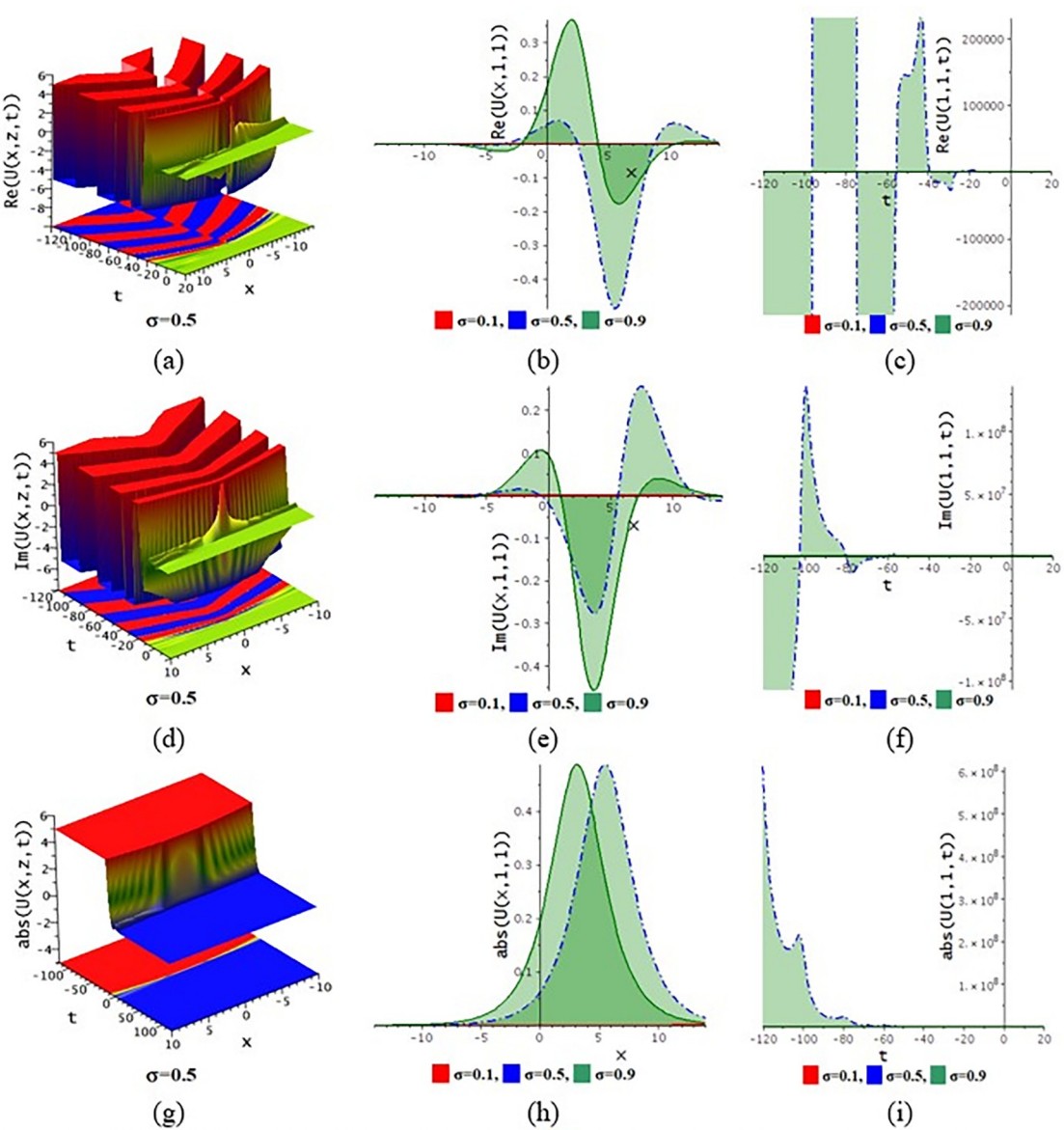

**Fig 11. 3D and 2D profile of the solution Eq (16) for the values** $\theta = -0.1, \varepsilon_3 = -1, z = 1, \varepsilon_2 = 0.2, \omega = 1, g_1 = -0.5, h_1 = -0.5, h_2 = -0.01, \varepsilon_1 = -1, n = 1.5.$

waves patterns. These solution provided the mechanisms of governing wave propagation and energy transfer within the material. In two dimensional plots, we show the effect of the fractional parameters.

**Fig 7** displays the 3D and 2D diagram of the Eq (13) for the values $\theta = -0.1, \varepsilon_3 = -1, z = 1, \varepsilon_2 = 0.2, \tau = 1, g_1 = -0.5, h_1 = -0.5, h_2 = -0.01, \varepsilon_1 = -1, n = 1.5.$ The real portion (**Fig 7A–7C**) and imaginary portion (**Fig 7D–7F**) show the interaction between periodic lump wave and periodic wave and the absolute (**Fig 7G–7I**) form of the solution represents the interaction of kink and periodic wave. In two dimensional diagrams, we show the fractional parameter effect. The solution Eq (13) visualizes the 3D and 2D diagram in **Fig 8A–8I** for the parameters

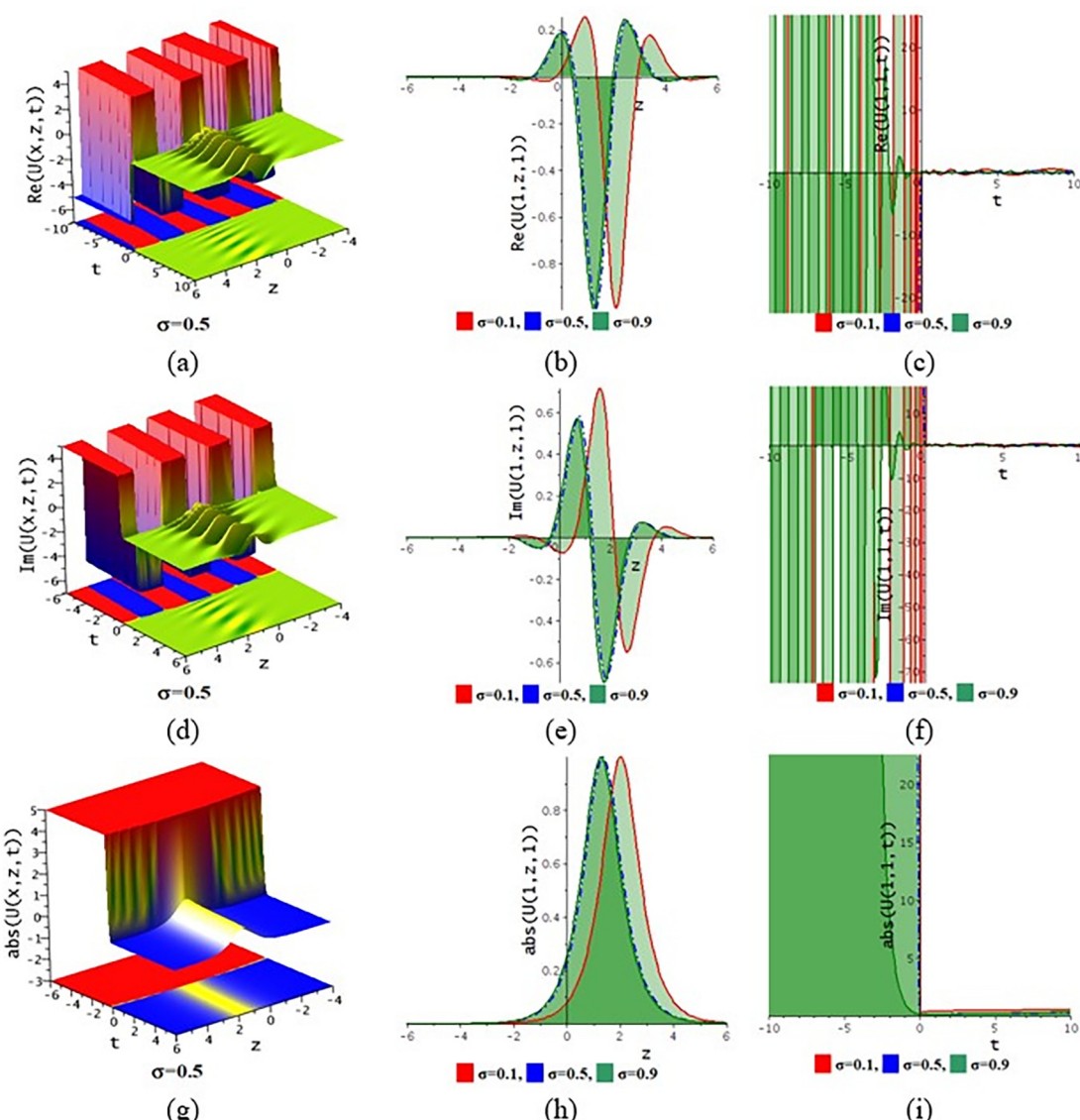

**Fig 12. 3D and 2D profile of the solution Eq (16) for the values** $\theta = -0.1, \varepsilon_3 = 1, x = 1, \varepsilon_2 = 1, \omega = 0.1, g_1 = 2, h_1 = 1, h_2 = 2, \varepsilon_1 = -0.1, n = 1.5.$

$\theta = -0.1, \varepsilon_3 = 1, z = 1, \varepsilon_2 = -0.2, \tau = 1, g_1 = 0.5, h_1 = -3, h_2 = -1.5, \varepsilon_1 = 1, n = 1.5.$ The real portion (**Fig 8A–8C**) and imaginary portion (**Fig 8D–8F**) show the double periodic wave and the absolute (**Fig 8G–8I**) form of the solution represents the interaction of soliton wave and kink wave. **Fig 9** displays the 3D and 2D diagram of the Eq (14) for the values $\theta = 0.1, \varepsilon_3 = 2, \varepsilon_2 = 0.1, \tau = -2, g_1 = 0.1, h_1 = 3, h_2 = 3, \varepsilon_1 = 1, n = 1.5.$ The real portion (**Fig 9A–9C**) and imaginary portion (**Fig 9D–9F**) show the periodic breather wave and the absolute (Fig 9g-9i) form of the solution represents the kinky-periodic wave. In two dimensional diagrams, we provides the fractional parameter effect. The solution Eq (14) visualizes the 3D and 2D diagram in **Fig 10** for the parameters $\theta = 0.1, z = 1, \varepsilon_3 = 2, \varepsilon_2 = 0.1, \tau = 0.5, g_1 = 0.1, h_1 = -0.5, h_2 = 3, \varepsilon_1 = 1, n = 1.5.$ The real portion (**Fig 10A–10C**) and imaginary portion (**Fig 10D–10F**) show the double periodic wave and the absolute (**Fig 10G–10I**)

form of the solution represents the interaction between soliton wave and kink wave. **Fig 11** displays the 3D and 2D diagram of the solution Eq (16) for the parameters $\theta = -0.1, \varepsilon_3 = -1, z = 1, \varepsilon_2 = 0.2, \omega = 1, g_1 = -0.5, h_1 = -0.5, h_2 = -0.01, \varepsilon_1 = -1, n = 1.5$. The real portion (**Fig 11A–11C**) and imaginary portion (**Fig 11D–11F**) show the periodic wave and the absolute (**Fig 11G–11I**) form of the solution represents the kink shape wave. In two dimensional diagrams, we provides the fractional parameter effect. The solution Eq (16) visualizes the 3D and 2D diagram in **Fig 12** for the parameters $\theta = -0.1, \varepsilon_3 = 1, x = 1, \varepsilon_2 = 1, \omega = 0.1, g_1 = 2, h_1 = 1, h_2 = 2, \varepsilon_1 = -0.1, n = 1.5$. The real portion (**Fig 12A-12C**) and imaginary portion (**Fig 12D-12F**) show the periodic wave with breather and the absolute (**Fig 12G-12I**) form of the solution characterises the interaction between anti-kink and soliton wave.

## 7. Conclusion

This study has been successfully focused on finding precise optical wave patterns within the field of optical physics, particularly examining the complex tMfPW equation through chaotic and sensitivity analysis. This equation is essential in optics as it helps explain optical phenomena, including solitons, nonlinear effects, and wave interactions. By using an EJEFE method, we derive various optical wave patterns characterized by trigonometric, and hyperbolic functions. We then employ principles from planar dynamical systems to explore the chaotic behaviors and sensitivity analysis inherent in the dynamical system. To confirm that small changes in initial conditions have minimal impact on the stability of the solution through chaotic behaviors. Maple software validates these results for accuracy. We also utilize dynamic visualizations, such as 2D, and 3D with density plots, to demonstrate different soliton patterns, including periodic waves, the interaction of periodic waves and lump waves, periodic breather waves, the interaction of kink and bell waves, kinky periodic waves, double periodic waves patterns. These visualizations offer insight into the intriguing behavior of optical phenomena. The solutions derived using this method highlight its effectiveness, reliability, and simplicity compared to other approaches.

## Supporting information

**S1 Data.**
(DOCX)

## Author Contributions

**Data curation:** M. M. Rahman.

**Formal analysis:** Md. Mamunur Roshid, Mohammad Safi Ullah, M. M. Rahman, Harun-Or-Roshid.

**Methodology:** Md. Mamunur Roshid, Mohammad Safi Ullah, M. M. Rahman, Harun-Or-Roshid.

**Software:** Harun-Or- Roshid.

**Supervision:** M. M. Rahman.

**Validation:** M. M. Rahman.

**Writing – original draft:** Md. Mamunur Roshid, Mohammad Safi Ullah.

**Writing – review & editing:** Md. Mamunur Roshid.

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
