## [Decision Letter · Decision Letter 0]

6 Aug 2024

PONE-D-24-28379Chaotic behavior, sensitivity analysis and Jacobian elliptic function solution of M-fractional paraxial wave with Kerr law nonlinearityPLOS ONE

Dear Dr. Roshid,

Thank you for submitting your manuscript to PLOS ONE. Reviews provided by three referees imply that the manuscript is definitely inappropriate for the publication in the present form. In particular, one review, presented by a top expert in the field, is outright negative. If you believe that you can thoroughly revise the paper and properly address critical comments, you have an option to resubmit the paper.

Please include the following items when submitting your revised manuscript:A rebuttal letter that responds to each point raised by the academic editor and reviewer(s). You should upload this letter as a separate file labeled 'Response to Reviewers'.A marked-up copy of your manuscript that highlights changes made to the original version. You should upload this as a separate file labeled 'Revised Manuscript with Track Changes'.An unmarked version of your revised paper without tracked changes. You should upload this as a separate file labeled 'Manuscript'.

Yours sincerely,

Boris Malomed

Academic Editor

PLOS ONE

Journal Requirements:

6. Please ensure that you refer to Figure 4 in your text as, if accepted, production will need this reference to link the reader to the figure.

Reviewers' comments:

Reviewer's Responses to Questions

**Comments to the Author**

1. Is the manuscript technically sound, and do the data support the conclusions?

Reviewer #1: Partly

Reviewer #2: Partly

Reviewer #3: Yes

2. Has the statistical analysis been performed appropriately and rigorously? 

Reviewer #1: No

Reviewer #2: N/A

Reviewer #3: Yes

3. Have the authors made all data underlying the findings in their manuscript fully available?

Reviewer #1: No

Reviewer #2: Yes

Reviewer #3: Yes

4. Is the manuscript presented in an intelligible fashion and written in standard English?

Reviewer #1: Yes

Reviewer #2: Yes

Reviewer #3: Yes

5. Review Comments to the Author

Reviewer #1: I have read this paper carefully. I see that the authors work very hard but unfortunately they have not found new results. My remarks on this paper are the following. 1) I believe that the authors have to give references on Eq. (1) where this equation was first derived. 2) After equation (1) the authors wrote that the expression in the y direction but I cannot see any function which depends on y direction. I see the only function R(x,z,t). 3) The Section (2.2) is devoted to consideration Eq. (2), but Eq. (2) is nonlinear ordinary differential equation. 4) Solution of Eq. (2) is well-known but the authors present a few of exact solutions. 5) The looked for of Equation (2) using formula (3) but they did not say what q_1, q_2 and p_1.The authors present different formulas for solution of Eq. (2) but these solutions are the same. More interesting Section 2 of paper. They present the consideration the chaotic nature. 6) However the approach by authors is well known as well. See , for example, the book “Chaotic Transitions in deterministic and Stochastic Dynamical Systems” by Emil Simiu, Prinston University Press. Taking into account the remarks above unfortinately I do not recommend this paper for publication.

Reviewer #2: Review of the article:

Chaotic behavior, sensitivity analysis and Jacobian elliptic function solution

of M-fractional paraxial wave with Kerr law nonlinearity.

Manuscript Number: PONE-D-24-28379

ESSENTIALS OF THIS ARTICLE:

In this paper the authors find several solutions of a fractional version of the 2D NLS (two-dimensional nonlinear Schrödinger) equation (which the authors call “paraxial wave equation”). In this article “x” is the evolution variable, and “t” and “z” are the “transversal” variables. The second-order time derivative has been replaced by an M-fractional derivative, which is a generalization of the conformable derivative.

Proposing a cleverly chosen similarity reduction, defined by the equations shown in (6), and the form of R(x,z,t) written above Eq. (6), the authors manage to reduce the fractional 2D NLS equation (1) to the ODE (ordinary differential equation) shown in (7). Then several particular solutions of this ODE are obtained by means of the EJEFE method.

Afterwards the authors study a perturbed form of Eq. (7) [shown in (17)], and obtain quasiperiodic and chaotic solutions of this perturbed equation.

OPINION OF THE ARTICLE:

The fractional 2D NLS equation studied in this article is an interesting equation, the solutions obtained are interesting, and the procedure to obtain these solutions is also interesting. The reduction of the fractional 2D NLS equation (1) to the ODE (7) is an excellent result (the most interesting result of the paper, from my point of view).

HOWEVER, the paper contains some errors and undefined symbols, which are probably the consequence of a hasty and careless writing. Moreover, some of the statements made in this article are, in my opinion, incorrect.

ISSUES THAT SHOULD BE CORRECTED:

ISSUE 1:

In the fourth line below Eq. (1) it is written “in the y and t directions”. This statement is wrong. It should say “in the t and z directions”.

ISSUE 2:

In pag. 3 of the paper the authors mention References [34]-[43].

But the list of references presented at the end of the article only contains 32 references!!!

ISSUE 3:

In the definition of the M-fractional derivative the meaning of the “t” with subscript kappa (which appears within the argument of the function “u”) is not explained. Moreover, in the right-hand-side (rhs) of the equation which defines this derivative, the parameter “M” does not appear. Please explain.

ISSUE 4:

Below the definition of the M-derivative, in the section “Features”, in the equations (a), (b), (c) and (f), the meaning of the letters with the subscript kappa (le letters l, m, u and v) is not explained.

ISSUE 5:

Two lines below Eq. (2) the authors say that:

“Balancing H’’ and H3, yields M=1”

but the authors have not said what is “M”. As I said in the Issue 3, the parameter M does not appear in the rhs of the equation which defines de M-derivative. So, it is incomprehensible how the equation M=1 is obtained.

ISSUE 6:

In the first line below Eq. (5) the authors mention the equations (6)-(8), but these equations have not been presented at this point of the article. Moreover, it is quite clear that in this line [below Eq. (5)], the authors are not referring to the Eqs. (6)-(8) which appear in Section 3.

ISSUE 7:

Two lines below Eq. (9) the equation M=1 appears again. As I said in Issue 5, we don´t know what is M, and therefore the origin of the equation M=1 is incomprehensible.

ISSUE 8:

Two lines above Eq. (17) the authors mention a “superficial component”. This term is a misleading, as it is not related to any “surface” at all. Moreover, as far as I could see, this term is never mentioned in Ref. [31].

ISSUE 9:

In the line above Eq. (17) the authors mention the “system Eq. (30)”. But the article contains 17 equations, consequently the mention of Eq. (30) is obviously incorrect.

ISSUE 10:

The numbering of the sections of this article is an absolute disaster, revealing an extremely careless work.

In page 3 the Sec. 2 begins.

In page 5, Sec. 3 begins.

Then, in page 6, another “Section 2” begins.

Then, in page 8, the Section 8 begins!.

And in page 9, the Section 5 begins!

Finally, in page 14, the Section 6 closes the paper!

ISSUE 11:

In page 6, in the Section entitled “Chaotic nature”, the authors modify their Eq. (7), introducing an oscillatory perturbation in time [the last term in the second equation shown in (17)]. And then they show that the solutions of this new ODE [Eq. (17)] exhibit a chaotic behavior. But the chaotic behavior of the solutions of Eq. (17) does not imply that Eq. (7) [or Eq. (1)] is a chaotic system.

The chaotic nature of a system (i.e., of an equation), is proved by perturbing THE INITIAL CONDITIONS USED, and NOT by perturbing the equation itself. Even though some authors may consider that an equation is “chaotic” if the solutions exhibit a huge change when the equation is perturbed, THIS IS NOT THE USUAL DEFINITION OF CHAOS. At least, it is not the definition of “chaos” used by Edward Lorenz (who was one of the founders of the theory of chaos).

Consequently, I consider that the authors must explain clearly that the chaotic solutions of Eq. (17) do not imply that the system described by Eq. (7) [or Eq. (1)] is chaotic.

RECOMMENDATION:

I consider that if the authors correct the eleven issues mentioned above, the article will be suitable for publication in PLOS ONE.

Reviewer #3: The authors investigated various types of wave solutions in the M-fractional paraxial wave equation with Kerr law nonlinearity, including periodic waves, lumpperiodic waves, periodic breather waves, kink-bell waves, kinky-periodic waves, anti-kinky-periodic waves, double-periodic waves. In addition, the chaotic phenomena are also studied in this work. The wave equations with M-fractional derivative attracted many attentions in recent years and the results presented in this work are interesting. However, some issues should be fixed, and my comments are as follows:

1. The physical scene (settings) for M-fractional derivative may be a good supplement to this work.

2. I am wondering if such M-fractional derivative can be expressed by other easier forms that can be solved numerically.

3. Where is the Eq. (30) the authors pointed out before Eq. (17) ?

4. I assume the references of [34] to [42] in the Introduction should be wrong. The authors should check it.

5. Some review should be added for the better understanding of fractional derivative.

[1] B. A. Malomed. Optical solitons and vortices in fractional media: a mini-review of recent results. Photonics 8(9), 353 (2021).

[2] B. A. Malomed. Basic fractional nonlinear-wave models and solitons. Chaos 34, 022102 (2024).

[3] D. Mihalache. Localized structures in optical media and Bose-Einstein condensates: An overview of recent theoretical and experimental results. Rom. Rep. Phys. 76, 402 (2024).

6. PLOS authors have the option to publish the peer review history of their article (what does this mean?). If published, this will include your full peer review and any attached files.

Reviewer #1: No

Reviewer #2: No

Reviewer #3: No

---

## [Author Response · Author response to Decision Letter 0]

21 Aug 2024

Reviewer Response

Reviewer #1:

I have read this paper carefully. I see that the authors work very hard but unfortunately they have not found new results. My remarks on this paper are the following. 

1) I believe that the authors have to give references on Eq. (1) where this equation was first derived. 

Response: The necessary correction is done in section 1 and we added the reference in [19]. 

2) After equation (1) the authors wrote that the expression in the y direction but I cannot see any function which depends on y direction. I see the only function R(x,z,t). 

Response: Thanks. The necessary correction is done.

3) The Section (2.2) is devoted to consideration Eq. (2), but Eq. (2) is nonlinear ordinary differential equation. 

Response: The necessary correction is done.

4) Solution of Eq. (2) is well-known but the authors present a few of exact solutions. 

Response: We appreciate the reviewer's comment. While it is true that the general solution to equation (2) is well-known, our contribution lies in presenting a few exact solutions in specific forms that may not be widely recognized or easily derived. These solutions provide additional insight into the behavior of the governing model under conditions, and we believe they add value to the existing body of knowledge.

5) The looked for of Equation (2) using formula (3) but they did not say what q_1, q_2, and p_1.The authors present different formulas for solution of Eq. (2) but these solutions are the same. More interesting Section 2 of paper. They present the consideration the chaotic nature. 

Response: The necessary correction is done in section 2.2 and 3.

6) However the approach by authors is well known as well. See , for example, the book “Chaotic Transitions in deterministic and Stochastic Dynamical Systems” by Emil Simiu, Prinston University Press. Taking into account the remarks above unfortinately I do not recommend this paper for publication.

Response: The reviewer’s mentioned paper described Melnikov's theory for chaotic dynamics. However, our paper uses a planar 

dynamical system to analyze the chaotic behavior of the governing model.

Reviewer #2: Review of the article:

Chaotic behavior, sensitivity analysis and Jacobian elliptic function solution

of M-fractional paraxial wave with Kerr law nonlinearity.

Manuscript Number: PONE-D-24-28379

ESSENTIALS OF THIS ARTICLE:

In this paper the authors find several solutions of a fractional version of the 2D NLS (two-dimensional nonlinear Schrödinger) equation (which the authors call “paraxial wave equation”). In this article “x” is the evolution variable, and “t” and “z” are the “transversal” variables. The second-order time derivative has been replaced by an M-fractional derivative, which is a generalization of the conformable derivative.

Proposing a cleverly chosen similarity reduction, defined by the equations shown in (6), and the form of R(x,z,t) written above Eq. (6), the authors manage to reduce the fractional 2D NLS equation (1) to the ODE (ordinary differential equation) shown in (7). Then several particular solutions of this ODE are obtained by means of the EJEFE method.

Afterwards the authors study a perturbed form of Eq. (7) [shown in (17)], and obtain quasiperiodic and chaotic solutions of this perturbed equation.

OPINION OF THE ARTICLE:

The fractional 2D NLS equation studied in this article is an interesting equation, the solutions obtained are interesting, and the procedure to obtain these solutions is also interesting. The reduction of the fractional 2D NLS equation (1) to the ODE (7) is an excellent result (the most interesting result of the paper, from my point of view).

HOWEVER, the paper contains some errors and undefined symbols, which are probably the consequence of a hasty and careless writing. Moreover, some of the statements made in this article are, in my opinion, incorrect.

ISSUES THAT SHOULD BE CORRECTED:

ISSUE 1:

In the fourth line below Eq. (1) it is written “in the y and t directions”. This statement is wrong. It should say “in the t and z directions”.

Response: The necessary correction is done.

ISSUE 2:

In pag. 3 of the paper the authors mention References [34]-[43].

But the list of references presented at the end of the article only contains 32 references!!!

Response: The necessary correction is done.

ISSUE 3:

In the definition of the M-fractional derivative the meaning of the “t” with subscript kappa (which appears within the argument of the function “u”) is not explained. Moreover, in the right-hand-side (rhs) of the equation which defines this derivative, the parameter “M” does not appear. Please explain.

Response: The necessary explanation is done in section 2.1 

ISSUE 4:

Below the definition of the M-derivative, in the section “Features”, in the equations (a), (b), (c) and (f), the meaning of the letters with the subscript kappa (le letters l, m, u and v) is not explained.

Response: The necessary correction is done.

ISSUE 5:

Two lines below Eq. (2) the authors say that:

“Balancing H’’ and H3, yields M=1”

but the authors have not said what is “M”. As I said in the Issue 3, the parameter M does not appear in the rhs of the equation which defines de M-derivative. So, it is incomprehensible how the equation M=1 is obtained.

Response: We explain it in section 2.2 and section 3.

ISSUE 6:

In the first line below Eq. (5) the authors mention the equations (6)-(8), but these equations have not been presented at this point of the article. Moreover, it is quite clear that in this line [below Eq. (5)], the authors are not referring to the Eqs. (6)-(8) which appear in Section 3.

Response: Thanks. The necessary correction is done.

ISSUE 7:

Two lines below Eq. (9) the equation M=1 appears again. As I said in Issue 5, we don´t know what is M, and therefore the origin of the equation M=1 is incomprehensible.

Response: we explain it in section 2.2.

ISSUE 8:

Two lines above Eq. (17) the authors mention a “superficial component”. This term is a misleading, as it is not related to any “surface” at all. Moreover, as far as I could see, this term is never mentioned in Ref. [31].

Response: We modified it.

ISSUE 9:

In the line above Eq. (17) the authors mention the “system Eq. (30)”. But the article contains 17 equations, consequently the mention of Eq. (30) is obviously incorrect.

Response: We modified it.

ISSUE 10:

The numbering of the sections of this article is an absolute disaster, revealing an extremely careless work.

In page 3 the Sec. 2 begins.

In page 5, Sec. 3 begins.

Then, in page 6, another “Section 2” begins.

Then, in page 8, the Section 8 begins!.

And in page 9, the Section 5 begins!

Finally, in page 14, the Section 6 closes the paper!

Response: We modified it.

ISSUE 11:

In page 6, in the Section entitled “Chaotic nature”, the authors modify their Eq. (7), introducing an oscillatory perturbation in time [the last term in the second equation shown in (17)]. And then they show that the solutions of this new ODE [Eq. (17)] exhibit a chaotic behavior. But the chaotic behavior of the solutions of Eq. (17) does not imply that Eq. (7) [or Eq. (1)] is a chaotic system.

The chaotic nature of a system (i.e., of an equation), is proved by perturbing THE INITIAL CONDITIONS USED, and NOT by perturbing the equation itself. Even though some authors may consider that an equation is “chaotic” if the solutions exhibit a huge change when the equation is perturbed, THIS IS NOT THE USUAL DEFINITION OF CHAOS. At least, it is not the definition of “chaos” used by Edward Lorenz (who was one of the founders of the theory of chaos).

Consequently, I consider that the authors must explain clearly that the chaotic solutions of Eq. (17) do not imply that the system described by Eq. (7) [or Eq. (1)] is chaotic.

Response: We include the following paragraph in the “Chaotic nature” section:

It is important to note that the chaotic behavior observed in the solutions to Eq. (17) arises due to the introduction of an oscillatory perturbation to the system. This does not necessarily imply that the original system described by Eq. (7) [or Eq. (1)] is chaotic in the traditional sense of chaos theory, where chaos is typically characterized by sensitivity to initial conditions rather than sensitivity to perturbations in the system itself. This distinction aligns with the definition of chaos introduced by Edward N. Lorenz and is crucial for accurately characterizing the system's dynamics [37].

[37] Lorenz EN. The Essence of Chaos. University of Washington Press. 1993; 181–206.

This clarification should help address the reviewer's concern while maintaining the integrity of your analysis.

Reviewer #3:

The authors investigated various types of wave solutions in the M-fractional paraxial wave equation with Kerr law nonlinearity, including periodic waves, lumpperiodic waves, periodic breather waves, kink-bell waves, kinky-periodic waves, anti-kinky-periodic waves, double-periodic waves. In addition, the chaotic phenomena are also studied in this work. The wave equations with M-fractional derivative attracted many attentions in recent years and the results presented in this work are interesting. However, some issues should be fixed, and my comments are as follows:

1. The physical scene (settings) for M-fractional derivative may be a good supplement to this work.

Response: Thanks for your response.

2. I am wondering if such M-fractional derivative can be expressed by other easier forms that can be solved numerically.

Response: Thanks

3. Where is the Eq. (30) the authors pointed out before Eq. (17) ?

Response: we modified it.

4. I assume the references of [34] to [42] in the Introduction should be wrong. The authors should check it.

Response: we modified it.

5. Some review should be added for the better understanding of fractional derivative.

Response: We cited some related article in [16, 17, 18]. 

6. PLOS authors have the option to publish the peer review history of their article (what does this mean?). If published, this will include your full peer review and any attached files.

---

## [Decision Letter · Decision Letter 1]

4 Sep 2024

PONE-D-24-28379R1Chaotic behavior, sensitivity analysis and Jacobian elliptic function solution of M-fractional paraxial wave with Kerr law nonlinearityPLOS ONE

Dear Dr. Roshid,

Thank you for resubmitting your manuscript to PLOS ONE. A new review suggests that an additional revision is necessary.

We look forward to receiving your revised manuscript.

Kind regards,

Boris Malomed

Academic Editor

PLOS ONE

Journal Requirements:

Reviewers' comments:

Reviewer's Responses to Questions

**Comments to the Author**

1. If the authors have adequately addressed your comments raised in a previous round of review and you feel that this manuscript is now acceptable for publication, you may indicate that here to bypass the “Comments to the Author” section, enter your conflict of interest statement in the “Confidential to Editor” section, and submit your "Accept" recommendation.

Reviewer #2: (No Response)

Reviewer #3: All comments have been addressed

2. Is the manuscript technically sound, and do the data support the conclusions?

Reviewer #2: Yes

Reviewer #3: Yes

3. Has the statistical analysis been performed appropriately and rigorously? 

Reviewer #2: N/A

Reviewer #3: Yes

4. Have the authors made all data underlying the findings in their manuscript fully available?

Reviewer #2: Yes

Reviewer #3: Yes

5. Is the manuscript presented in an intelligible fashion and written in standard English?

Reviewer #2: Yes

Reviewer #3: Yes

6. Review Comments to the Author

Reviewer #2: Review of the revised version of the article:

Chaotic behavior, sensitivity analysis and Jacobian elliptic function solution

of M-fractional paraxial wave with Kerr law nonlinearity.

Manuscript Number: PONE-D-24-28379R1

In the revised version of this article the authors have taken into account the observations mentioned in my previous review. However, two of the issues mentioned in that review have not been answered satisfactorily: Issues 3 and 4.

In page 4 the authors added a new paragraph entitled “Features”, aimed to clarify the Issues 3 and 4. However, the “explanations” contained in this paragraph are completely unsatisfactory. The statement:

“M indicates the truncation point or order”

is absolutely ambiguous, and it does noy clarify how the parameter M enters in the calculation of the limit which appears in the rhs (right hand side) of the equation which defines the M-derivative. Additionally, the statement:

“kappa represents a scaling factor or a characteristic constant associated with the operator”

is also imprecise and ambiguous, as the letter “kappa” enters in the rhs of the definition of the M-derivative as the name of a subindex, and it does not occupy the position of a numerical factor or a constant. Therefore, it is necessary to explain with absolute clarity how the symbol t_kappa depends on kappa.

And, in a similar way, the authors do not explain how the parameters l_kappa, m_kappa, u_kappa and v_kappa, which appear in the equations (a), (b) and (c) presented in page 4, depend on kappa. Therefore, it is necessary to explain with absolute clarity how these four symbols depend on kappa.

Recommendation:

As the essential novelty of the equation studied in this article is the use of the M-derivative, the definition and the properties of this operator must be absolutely clear in the article. Therefore, in my opinion, this article should not be accepted for publication in PLOS ONE until these issues are presented with absolute clarity.

Reviewer #3: The authors have replied all my comments. This manuscript looks much better now. I have no further comments on this work.

7. PLOS authors have the option to publish the peer review history of their article (what does this mean?). If published, this will include your full peer review and any attached files.

Reviewer #2: No

Reviewer #3: No

---

## [Author Response · Author response to Decision Letter 1]

8 Nov 2024

Reviewer #2: Review of the revised version of the article:

Chaotic behavior, sensitivity analysis and Jacobian elliptic function solution

of M-fractional paraxial wave with Kerr law nonlinearity.

Manuscript Number: PONE-D-24-28379R1

In the revised version of this article the authors have taken into account the observations mentioned in my previous review. However, two of the issues mentioned in that review have not been answered satisfactorily: Issues 3 and 4.

In page 4 the authors added a new paragraph entitled “Features”, aimed to clarify the Issues 3 and 4. However, the “explanations” contained in this paragraph are completely unsatisfactory. The statement:

“M indicates the truncation point or order”

is absolutely ambiguous, and it does noy clarify how the parameter M enters in the calculation of the limit which appears in the rhs (right hand side) of the equation which defines the M-derivative. Additionally, the statement:

“kappa represents a scaling factor or a characteristic constant associated with the operator”

is also imprecise and ambiguous, as the letter “kappa” enters in the rhs of the definition of the M-derivative as the name of a subindex, and it does not occupy the position of a numerical factor or a constant. Therefore, it is necessary to explain with absolute clarity how the symbol t_kappa depends on kappa.

And, in a similar way, the authors do not explain how the parameters l_kappa, m_kappa, u_kappa and v_kappa, which appear in the equations (a), (b) and (c) presented in page 4, depend on kappa. Therefore, it is necessary to explain with absolute clarity how these four symbols depend on kappa.

Ans.: In this operator(_κ^ )D_(M,t)^(σ,ϕ) , κ is real constant, known as a scaling factor or a characteristic constant associated with the operator. It is related to the operator through the truncated Mittag-Leffler function with one parameter, as discussed in [40]

(_κ^ )E_ϕ (t)=∑_(n=0)^κ▒t^n/Γ(ϕn + 1) .

In the feature, u=u(t),v=v(t) are the function of time t. We justify the properties of truncated M-fractional operator.

The parameter 𝑀 is used to denote that the function to be derived involves a Mittag-Leffler function with one parameter. Further discussion can be found in [41].

〖(_κ^ )D_(M,t)^(σ,ϕ) 〗^ u(t)=lim┬(ϵ→0)⁡〖(u(t_K E_ϕ (ϵt^(-σ) ))-u(t))/ϵ〗,t>0,ϕ>0.

Here, (_κ^ )E_ϕ (.) is a truncated Mittag-Leffler function of one parameter. In left side M covert into (_κ^ )E_ϕ (.) In right side. (See [41] (definition 2.1, 2.2))

Recommendation:

As the essential novelty of the equation studied in this article is the use of the M-derivative, the definition and the properties of this operator must be absolutely clear in the article. Therefore, in my opinion, this article should not be accepted for publication in PLOS ONE until these issues are presented with absolute clarity.

Reviewer #3: The authors have replied all my comments. This manuscript looks much better now. I have no further comments on this work.

Thanks for your recommends.

---

## [Editor Report · Decision Letter 2]

15 Nov 2024

Chaotic behavior, sensitivity analysis and Jacobian elliptic function solution of M-fractional paraxial wave with Kerr law nonlinearity

PONE-D-24-28379R2

Dear Dr. Roshid,

We’re pleased to inform you that your manuscript has been judged scientifically suitable for publication and will be formally accepted for publication once it meets all outstanding technical requirements.

Kind regards,

Boris Malomed

Academic Editor

PLOS ONE

Additional Editor Comments (optional):

Comments from PLOS Editorial Office: We note that one or more reviewers has recommended that you cite specific previously published works in an earlier round of revision. As always, we recommend that you please review and evaluate the requested works to determine whether they are relevant and should be cited. It is not a requirement to cite these works and you may remove them before the manuscript proceeds to publication. We appreciate your attention to this request.
---

## [Editor Report · Acceptance letter]

29 Nov 2024

PONE-D-24-28379R2 

PLOS ONE

Dear Dr. Roshid, 

I'm pleased to inform you that your manuscript has been deemed suitable for publication in PLOS ONE. Congratulations! Your manuscript is now being handed over to our production team.

Kind regards, 

on behalf of

Prof. Boris Malomed 

Academic Editor

PLOS ONE